# ENABLING COUNTERFACTUAL SURVIVAL ANALYSIS WITH BALANCED REPRESENTATIONS

## ABSTRACT

Balanced representation learning methods have been applied successfully to counterfactual inference from observational data. However, approaches that account for survival outcomes are relatively limited. Survival data are frequently encountered across diverse medical applications, *i.e.*, drug development, risk profiling, and clinical trials, and such data are also relevant in fields like manufacturing (for equipment monitoring). When the outcome of interest is time-to-event, special precautions for handling censored events need to be taken, as ignoring censored outcomes may lead to biased estimates. We propose a theoretically grounded unified framework for counterfactual inference applicable to survival outcomes. Further, we formulate a nonparametric hazard ratio metric for evaluating average and individualized treatment effects. Experimental results on real-world and semi-synthetic datasets, the latter which we introduce, demonstrate that the proposed approach significantly outperforms competitive alternatives in both survival-outcome predictions and treatment-effect estimation.

## 1 INTRODUCTION

Survival analysis or time-to-event studies focus on modeling the time of a future event, such as death or failure, and investigate its relationship with covariates or predictors of interest. Specifically, we may be interested in the *causal effect* of a given intervention or treatment on survival time. A typical question may be: will a given therapy increase the chances of survival of an individual or population? Such causal inquiries on survival outcomes are common in the fields of epidemiology and medicine (Robins, 1986; Hammer et al., 1996; Yusuf et al., 2016). As an important current example, the COVID-19 pandemic is creating a demand for methodological development to address such questions, specifically, when evaluating the effectiveness of a potential vaccine or therapeutic outside randomized controlled trial settings.

Traditional causal survival analysis is typically carried out in the context of a randomized controlled trial (RCT), where the treatment assignment is controlled by researchers. Though they are the gold standard for causal inference, RCTs are usually long-term engagements, expensive and limited in sample size. Alternatively, the availability of *observational* data with comprehensive information about patients, such as electronic health records (EHRs), constitutes a more accessible but also more challenging source for estimating causal effects (Häyrinen et al., 2008; Jha et al., 2009). Such observational data may be used to augment and verify an RCT, after a particular treatment is approved and in use (Gombar et al., 2019; Frankovich et al., 2011; Longhurst et al., 2014). Moreover, the wealth of information from observational data also allows for the estimation of the individualized treatment effect (ITE), namely, the causal effect of an intervention at the individual level. In this work, we develop a novel framework for *counterfactual time-to-event prediction* to estimate the ITE for survival or time-to-event outcomes from observational data.

Estimating the causal effect for survival outcomes in observational data manifests two principal challenges. First, the treatment assignment mechanism is not known *a priori*. Therefore, there may be variables, known as *confounders*, affecting both the treatment and survival time, which lead to selection bias (Bareinboim & Pearl, 2012), *i.e.*, that the distributions across treatment groups are not the same. In this work, we focus on selection biases due to confounding, but other sources may also be considered. For instance, patients who are severely ill are likely to receive more aggressive therapy, however, their health status may *also* inevitably influence survival. Traditional survival analysis

neglects such bias, leading to incorrect causal estimation. Second, the exact time-to-event is not always observed, *i.e.*, sometimes we only know that an event has *not* occurred up to a certain point in time. This is known as the *censoring* problem. Moreover, censoring might be informative depending on the characteristics of the individuals and their treatment assignments, thus proper adjustment is required for accurate causal estimation (Cole & Hernán, 2004; Díaz, 2019).

Traditional causal survival-analysis approaches typically model the effect of the treatment or covariates (not time or survival) in a parametric manner. Two commonly used models are the Cox proportional hazards (CoxPH) model (Cox, 1972) and the accelerated failure time (AFT) model (Wei, 1992), which presume a linear relationship between the covariates and survival probability. Further, proper weighting for each individual has been employed to account for confounding bias from these models (Austin, 2007; 2014; Hernán et al., 2005). For instance, probability weighting schemes that account for both selection bias and covariate dependent censoring have been considered for adjusted survival curves (Cole & Hernán, 2004; Díaz, 2019). Moreover, such probability weighting schemes have been applied to causal survival-analysis under time-varying treatment and confounding (Robins, 1986; Hernán et al., 2000). See van der Laan & Robins (2003); Tsiatis (2007); Van der Laan & Rose (2011); Hernán & Robins (2020) for an overview. Such linear specification makes these models interpretable but compromises their flexibility, and makes it difficult to adapt them for high-dimensional data or to capture complex interactions among covariates. Importantly, these methods lack a counterfactual prediction mechanism, which is key for ITE estimation (see Section 2).

Fortunately, recent advances in machine learning, such as representation learning or generative modeling, have enabled causal inference methods to handle high-dimensional data and to characterize complex interactions effectively. For instance, there has been recent interest in tree-based (Chipman et al., 2010; Wager & Athey, 2018) and neural-network-based (Shalit et al., 2017; Zhang et al., 2020) approaches. For pre-specified time-horizons, the nonparametric Random Survival Forest (RSF) (Ishwaran et al., 2008) and Bayesian Additive regression trees (BART) (Chipman et al., 2010) have been extended to causal survival analysis. RSF has been applied to causal survival forests with weighted bootstrap inference (Shen et al., 2018; Cui et al., 2020) while a BART is extended to account for survival outcomes in Surv-BART (Sparapani et al., 2016), and AFT-BART (Henderson et al., 2020). See Hu et al. (2020) for an extensive investigation of the causal survival tree-based methods. Alternatively, when estimating the ITE, neural-network-based methods propose to regularize the transformed covariates or representations for an individual to have balanced distributions across treatment groups, thus accounting for the confounding bias and improving ITE prediction. However, most approaches employing *representation learning* techniques for counterfactual inference deal with continuous or binary outcomes, instead of time-to-event outcomes with censoring (*informative or non-informative*). Hence, a principled generalization to the context of *counterfactual survival analysis* is needed.

In this work we leverage balanced (latent) representation learning to estimate ITE via counterfactual prediction of survival outcomes in observational studies. We develop a framework to predict event times from a low-dimensional transformation of the original covariate space. To address the specific challenges associated with counterfactual survival analysis, we make the following contributions:

- We develop an optimization objective incorporating adjustments for informative censoring, as well as a balanced regularization term bounding the generalization error for ITE prediction. For the latter, we repurpose a recently proposed bound (Shalit et al., 2017) for our time-to-event scenario.
- We propose a generative model for event times to relax restrictive survival linear and parametric assumptions, thus allowing for more flexible modeling. Our approach can also provide nonparametric uncertainty quantification for ITE predictions.
- We provide survival-specific evaluation metrics, including a new *nonparametric hazard ratio* estimator, and discuss how to perform model selection for survival outcomes. The proposed model demonstrates superior performance relative to the commonly used baselines in real-world and semi-synthetic datasets.
- We introduce a survival-specific semi-synthetic dataset and demonstrate an approach for leveraging prior randomized experiments in longitudinal studies for model validation.

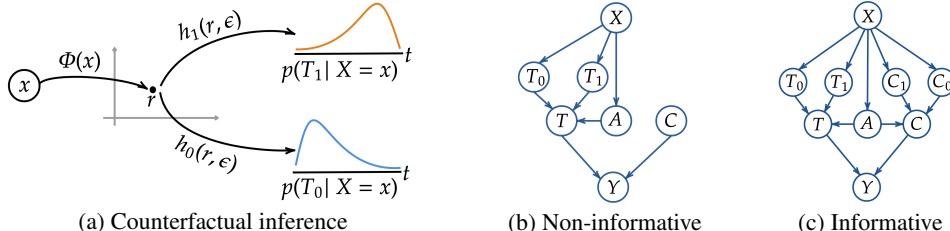

(a) Counterfactual inference      (b) Non-informative      (c) Informative

Figure 1: (a) Illustration of the proposed counterfactual survival analysis (CSA). Covariates $X = x$ are mapped into latent representation $r$ via deterministic mapping $r = \Phi(x)$. The potential outcomes are sampled from $t_a \sim p(T_A|X = x)$ for $A = a$ via stochastic mapping $h_A(r, \tilde{\epsilon})$, where randomness is induced with a flow-based transformation, $\tilde{\epsilon}$, of a simple distribution $p(\epsilon)$, *i.e.*, uniform or Gaussian. (b) and (c) show the proposed causal graphs for non-informative and informative censoring, respectively.

## 2   PROBLEM FORMULATION

We first introduce the basic setup for performing causal survival analysis in observational studies. Suppose we have $N$ units, with $N_1$ units being *treated* and $N_0$ in the *control* group ($N = N_1 + N_0$). For each unit (individual), we have covariates $X$, which can be heterogeneous, *e.g.*, a mixture of categorical and continuous covariates which, in the context of medicine, may include labs, vitals, procedure codes, *etc*. We also have a *treatment* indicator $A$, where $A = 0$ for the controls and $A = 1$ for the treated, as well as the outcome (event) of interest $T$. Under the potential-outcomes framework (Rubin, 2005), let $T_0$ and $T_1$ be the potential event times for a given subject under control and treatment, respectively. In practice we only observe one realization of the potential outcomes, *i.e.*, the *factual* outcome $T = T_A$, while the *counterfactual* outcome $T_{1-A}$ is unobserved.

In survival analysis, the problem becomes more difficult because we do *not* always observe the exact event time for each individual, but rather the time up to which we are certain that the event has not occurred; specifically, we have a (right) censoring problem, most likely due to the loss of follow-up. We denote the censoring time as $C$ and censoring indicator as $\delta \in \{0, 1\}$. The actual *observed time* is $Y = \min(T_A, C)$, *i.e.*, the outcome is observed (non-censored) if $T_A < C$ and $\delta = 1$.

In this work, we are interested in the expected difference between the $T_1$ and $T_0$ conditioned on $X$ for a given unit (individual), which is commonly known as the *individualized treatment effect* (ITE). Specifically, we wish to perform inference on the conditional distributions of $T_1$ and $T_0$, *i.e.*, $p(T_1|X)$ and $p(T_0|X)$, respectively, as shown in Figure 1a. In practice, we observe $N$ realizations of $(Y, \delta, X, A)$ for observed time, censoring indicator, covariates and treatment indicator, respectively; hence, from an observational study the dataset takes the form $\mathcal{D} = \{(y_i, \delta_i, x_i, a_i)\}_{i=1}^N$. Below, we discuss several common choices of estimands in survival analysis.

**Estimands of Interest**   We begin by considering survival analysis in the *absence* of an intervening treatment choice, $A$. Let $F(t|x) \triangleq P(T \le t|X = x)$ be the cumulative distribution function of the event (failure) time, $t$, given a realization of the covariates, $x$. Survival analysis is primarily concerned with characterization of the *survival function* conditioned on covariates $S(t|x) \triangleq 1 - F(t|x)$, and the *hazard function* or *risk score*, $\lambda(t|x)$, defined below. $S(t|x)$ is a monotonically decreasing function indicating the probability of survival up to time $t$. The hazard function measures the instantaneous probability of the event occurring between $\{t, t + \Delta t\}$ given $T > t$ and $\Delta t \to 0$. From standard definitions (Kleinbaum & Klein, 2010), the relationship between cumulative and hazard function is formulated as

$$\lambda(t|x) = \lim_{dt \to 0} \frac{P(t < T < t + dt|X = x)}{P(T > t|X = x)dt} = -\frac{d \log S(t|x)}{dt} = \frac{f(t|x)}{S(t|x)}. \tag{1}$$

From (1) we see that $f(t|x) \triangleq P(T = t|X = x) = \lambda(t|x)S(t|x)$, is the conditional *event time density function* (Kleinbaum & Klein, 2010).

Given the binary treatment $A$, we are interested in its impact on the survival time. For ITE estimation, we are also interested in the difference between the two potential outcomes $T_1, T_0$. Let $S_A(t|x)$ and $\lambda_A(t|x)$ denote the survival and hazard functions for the potential outcomes $T_A$, *i.e.*, $T_1$ and $T_0$. Several common estimands of interest include (Zhao et al., 2012; Trinquart et al., 2016): *difference*

*in expected lifetime*: $\text{ITE}(t, x) = \int_0^{t_{\max}} \{S_1(t|x) - S_0(t|x)\} \mathrm{d}t = \mathbb{E}\{T_1 - T_0|X = x\}$, *difference in survival function*: $\text{ITE}(t, x) = S_1(t|x) - S_0(t|x)$, and *hazard ratio*: $\text{ITE}(t, x) = \lambda_1(t|x)/\lambda_0(t|x)$. The inference difficulties associated with the above estimands from observational data are two-fold. First, there are confounders affecting both the treatment assignment and outcomes, which stem from selection bias, *i.e.*, the treatment and control covariate distributions are not necessarily the same. Also, we do not have direct knowledge of the conditional treatment assignment mechanism, *i.e.*, $P(A = a|X = x)$, also known as the *propensity score*. Let $\perp\!\!\!\perp$ denote statistical independence. For estimands to be identifiable from observational data, we make two assumptions: ($i$) $\{T_1, T_0\} \perp\!\!\!\perp A|X$, *i.e.*, no unobserved confounders or *ignorability*, and ($ii$) *overlap* in the covariate support $0 < P(A = 1|X = x) < 1$ *almost surely if* $p(X = x) > 0$. Second, the censoring mechanism is also unknown and may lead to bias without proper adjustment. We consider two censoring mechanisms in our work, ($i$) conditionally independent or *informative censoring*: $T \perp\!\!\!\perp C|X, A$, and ($ii$) random or *non-informative censoring*: $T \perp\!\!\!\perp C$. Note that for informative censoring, we also have to consider potential censoring times $C_1$ and $C_0$ and their conditionals $p(C_1|X)$ and $p(C_0|X)$, respectively. Figure 1 shows causal graphs illustrating these modeling assumptions.

## 3 MODELING

To overcome the above challenges and adjust for observational biases, we propose a unified framework for *counterfactual survival analysis* (CSA). Specifically, we repurpose the counterfactual bound in Shalit et al. (2017) for our time-to-event scenario and introduce a nonparametric approach for stochastic survival outcome predictions. Below we formulate a theoretically grounded and unified approach for estimating ($i$) the encoder function $r = \Phi(x)$, which deterministically maps covariates $x$ to their corresponding latent representation $r \in \mathbb{R}^d$, and ($ii$) two stochastic time-to-event generative functions, $h_A(\cdot)$, to implicitly draw samples from both potential outcome conditionals $t_a \sim p_{h,\Phi}(T_A|X = x)$, for $A = \{1, 0\}$, and where $t_a$ indicates the sample from $p_{h,\Phi}(T_A|X = x)$ is for $A = a$. Further, we formulate a general extension that accounts for informative censoring by introducing two stochastic censoring generative functions, $\nu_A(\cdot)$, to draw samples for potential censoring times $c_a \sim p_{\nu,\Phi}(C_A|X = x)$. The model-specifying functions, $\{h_A(\cdot), \nu_A(\cdot), \Phi(\cdot)\}$, are parameterized via neural networks. See the Supplementary Material (SM) for details. Figure 1a summarizes our modeling approach.

**Accounting for selection bias** We wish to estimate the potential outcomes, *i.e.*, event times, which are sampled by distributions parameterized by functions $\{h_A(\cdot), \Phi(\cdot)\}$, *i.e.*,

$$t \sim p_{h,\Phi}(T|X = x, A = a) \tag{2}$$

$$t_a \sim p_{h,\Phi}(T_a|X = x) \tag{3}$$

We obtain (3) from (2) via the *strong ignorability* assumption, *i.e.*, $\{T_0, T_1\} \perp\!\!\!\perp A|X$ (consistent with the causal graphs in Figure 1b and 1c) and $0 < P(A = a|X = x) < 1$, and the *consistency* assumption, *i.e.*, $T = T_A|A = a$. A similar argument can be made for informative censoring based on Figure 1c, so we can also write $c_a \sim p_{\nu,\Phi}(C_A|X = x)$. Given (3), model functions $\{h_A(\cdot), \Phi(\cdot)\}$ and $\nu_A(\cdot)$ for informative censoring can be learned by leveraging standard statistical optimization approaches, that minimize a loss hypothesis $\mathcal{L}$ given samples from the empirical distribution $(y, \delta, x, a) \sim p(Y, \delta, X, A)$, *i.e.*, from dataset $\mathcal{D}$. Specifically, we write $\mathcal{L} = \mathbb{E}_{(y,\delta,x,a) \sim p(Y,\delta,X,A)}[\ell_{h,\Phi}(t_a, y, \delta)]$, where $\ell_{h,\Phi}(t_a, y, \delta)$ is a loss function that measures the agreement of $t_a \sim p_{h,\Phi}(T_A|X = x)$ (and $c_a \sim p_{\nu,\Phi}(C_A|X = x)$ for informative censoring) with ground truth $\{y, \delta\}$, the observed time and censoring indicator, respectively.

For some parametric formulations of event time distribution $p_{h,\Phi}(T_A|X = x)$, *e.g.*, exponential, Weibull, log-Normal, *etc.*, and provided the censoring mechanism is non-informative, $-\ell_{h,\Phi}(t_a, y, \delta)$ is the closed form log likelihood. Specifically, $-\ell_{h,\Phi}(t_a, y, \delta) \triangleq \log p_{h,\Phi}(T_a|X = x) = \delta \cdot \log f_{h,\Phi}(t_a|x) + (1 - \delta) \cdot \log S_{h,\Phi}(t_a|x)$, which implies that the conditional event time density and survival functions can be calculated in closed form from transformations $\{h_A(\cdot), \Phi(\cdot)\}$ of $x$. See the SM for parametric examples of $\mathcal{L}$ accounting for informative censoring.

We further define the expected loss for a given realization of covariates $x$ and treatment assignment $a$ over observed times $y$ (censored and non-censored), and the censoring indicator $\delta$ as $\zeta_{h,\Phi}(x, a) \triangleq \mathbb{E}_{(y,\delta,x) \sim p(Y,\delta|X)} \ell_{h,\Phi}(t_a, y, \delta)$ as in Shalit et al. (2017). For a given subject with covariates $x$ and treatment assignment $a$, we wish to minimize both the factual and counterfactual losses, $\mathcal{L}_{\text{F}}$ and $\mathcal{L}_{\text{CF}}$,

respectively, by decomposing $\mathcal{L} = \mathcal{L}_\mathrm{F} + \mathcal{L}_\mathrm{CF}$ as follows

$$\mathcal{L}_\mathrm{F} = \mathbb{E}_{(x,a)\sim p(A,X)} \zeta_{h,\Phi}(x,a)\,, \quad \mathcal{L}_\mathrm{CF} = \mathbb{E}_{(x,a)\sim p(1-A,X)} \zeta_{h,\Phi}(x,a)\,. \qquad (4)$$

Let $u \triangleq P(A = 1)$ denote the marginal probability of treatment assignment. We can readily decompose the losses in (4) according to treatment assignments. The decomposed factual $\mathcal{L}_\mathrm{F} = u \cdot \mathcal{L}_\mathrm{F}^{A=1} + (1 - u) \cdot \mathcal{L}_\mathrm{F}^{A=0}$, and similarly, the decomposed counterfactual $\mathcal{L}_\mathrm{CF} = (1 - u) \cdot \mathcal{L}_\mathrm{CF}^{A=1} + u \cdot \mathcal{L}_\mathrm{CF}^{A=0}$. In practice, only *factual* outcomes are observed, hence, for a non-randomized non-controlled experiment, we cannot obtain an unbiased estimate of $\mathcal{L}_\mathrm{CF}$ from data due to selection bias (or confounding). Therefore, we bound $\mathcal{L}_\mathrm{CF}$ and $\mathcal{L}$ below following Shalit et al. (2017).

**Corollary 1** *Assume $\Phi(\cdot)$ is an invertible map, and $\alpha^{-1}\zeta_{h,\Phi}(x,a) \in G$, where $G$ is a family of functions, $p_\Phi^{A=a} \triangleq p_\Phi(R|A = a)$ is the latent distribution for group $A = a$, and $\alpha > 0$ is a constant. Then, we have:*

$$\mathcal{L}_\mathrm{CF} \leq (1 - u) \cdot \mathcal{L}_\mathrm{F}^{A=1} + u \cdot \mathcal{L}_\mathrm{F}^{A=0} + \alpha \cdot \mathrm{IPM}_G(p_\Phi^{A=1}, p_\Phi^{A=0})$$
$$\mathcal{L} \leq \mathcal{L}_\mathrm{F}^{A=1} + \mathcal{L}_\mathrm{F}^{A=0} + \alpha \cdot \mathrm{IPM}_G(p_\Phi^{A=1}, p_\Phi^{A=0})\,. \qquad (5)$$

The integral probability metric (IPM) (Müller, 1997; Sriperumbudur et al., 2012) measures the distance between two probability distributions $p$ and $q$ defined over $M$, *i.e.*, the latent space of $R$. Formally, $\mathrm{IPM}_G(p, q) \triangleq \sup_{g \in G} |\int_M g(m)(p(m) - q(m))\, dm|$, where $g : m \to \mathbb{R}$, represents a class of real-valued bounded measurable functions on $M$ (Shalit et al., 2017). Therefore, model functions $\{h_a(\cdot), \Phi(\cdot)\}$ can be learned by minimizing the upper bound in (5) consisting of $(i)$ only *factual* losses under both treatment assignments and $(ii)$ an IPM regularizer enforcing latent distributional equivalence between the treatment groups. Note that if the data originates from a RCT it follows (by construction) that $\mathrm{IPM}_G(p_\Phi^{A=1}, p_\Phi^{A=0}) = 0$.

**Accounting for censoring bias** Below we formulate an approach for estimating functions $h_A(\cdot)$ and $\nu_A(\cdot)$ for synthesizing (sampling) non-censored $t_a \sim p_{h,\Phi}(T_A|X = x)$ and censored $c_a \sim p_{\nu,\Phi}(C_A|X = x)$ times, respectively. While some parametric assumptions for $p_{h,\Phi}(T_A|X = x)$ yield easy-to-evaluate closed forms for $S_{h,\Phi}(t_a|x)$ that can be used as likelihood for censored observations, they are restrictive, and have been shown to generate unrealistic high variance samples (Chapfuwa et al., 2018). So motivated, we seek a nonparametric likelihood-based approach that can model a flexible family of distributions, with an easy-to-sample approach for event times $t_a \sim p_{h,\Phi}(T_a|X = x)$. We model the event time generation process with a source of randomness, $p(\epsilon)$, *e.g.* Gaussian or uniform, which is obtained from a neural-network-based nonlinear transformation. In the experiments we use a *planar flow* formulation parameterized by $\{U_h, W_h, b_h\}$ (Rezende & Mohamed, 2015), however, other specifications can also be used. Note that Miscouridou et al. (2018) has previously leveraged normalizing flows for survival analysis, however, our approach is very different in that it focuses on formulating $i$) a counterfactual survival analysis framework that accounts for *informative or non-informative* censoring mechanisms and confounding, and $ii$) model event times as a continuous variable instead of discretizing them. Specifically, we transform the source of randomness, $\epsilon$, using a single layer specification as follows

$$\tilde{\epsilon}_h = \epsilon + U_h \tanh(W_h \epsilon + b_h)\,, \quad \epsilon \sim \mathrm{Uniform}(0, 1)\,, t_a = h_A(r, \tilde{\epsilon}_h)\,, \quad r = \Phi(x) \qquad (6)$$

where $\{U_h, W_h\} \in \mathbb{R}^{d \times d}$, $\{b_h, \epsilon\} \in \mathbb{R}^d$, $d$ is the dimensionality of the normalizing flow; each component of $\epsilon$ is drawn independently from $\mathrm{Uniform}(0, 1)$, and $\tilde{\epsilon}_h$ may be viewed as a skip connection with stochasticity in $\epsilon$. Further, $h_A(r, \tilde{\epsilon}_h)$ and $\Phi(x)$ are time-to-event generative and encoding functions, respectively, parameterized as neural networks. For simplicity, the dimensions of $r$ and $\epsilon$ are set to $d$, however, they can be set independently if desired. In practice, we are interested in generating realistic event-time samples; therefore, we account for both censored and non-censored observations by adopting the objective from Chapfuwa et al. (2018), formulated as

$$\mathcal{L}_\mathrm{F}^\mathrm{CSA} \triangleq \mathbb{E}_{(y,\delta,x,a)\sim p(Y,\delta,X,A),\epsilon\sim p(\epsilon)} [\delta \cdot (|y - t_a|) + (1 - \delta) \cdot (\max(0, y - t_a))]\,, \qquad (7)$$

where the first term encourages sampled event times $t_a$ to be close to $y$, the ground truth for observed events, *i.e.*, $\delta = 1$, while penalizing $t_a$ for being smaller than the censoring time when $\delta = 0$. Further, the expectation is taken over samples (a minibatch) from empirical distribution $p(Y, \delta, X, A)$.

**Informative censoring**   We model informative censoring similar to (7) but mirroring the censoring indicators to encourage accurate censoring time samples $c_a$ for $\delta = 0$, while penalizing $c_a$ for being smaller than $y$ for $\delta = 1$ (observed events). Specifically, we set an independent source of randomness like in (6) but parameterized by $\{U_\nu, W_\nu, b_\nu\}$ and censoring generative functions $\nu_A(r, \tilde{\epsilon}_\nu)$, parameterized as neural networks, where $c_a \sim p_{\nu,\Phi}(C_A | X = x)$ formulated as

$$\ell_c(\nu, \Phi) = \mathbb{E}_{(y,\delta,x,a)\sim p(y,\delta,X,A),\epsilon\sim p(\epsilon)} \left[ (1-\delta) \cdot (|y - c_a|) + \delta \cdot (\max(0, y - c_a)) \right] . \quad (8)$$

Further, we introduce an additional time-order-consistency loss that enforces the correct order of the observed time relative to the censoring indicator, *i.e.*, $c_a < t_a$ if $\delta = 0$ and $t_a < c_a$ if $\delta = 1$, thus

$$\ell_{\text{TC}}(h, \nu, \Phi) = \mathbb{E}_{(\delta,x,a)\sim p(\delta,X,A),\epsilon\sim p(\epsilon)} \left[ \delta \cdot (\max(0, t_a - c_a)) + (1-\delta) \cdot (\max(0, c_a - t_a)) \right] \quad (9)$$

Note that $\ell_{\text{TC}}(h, \nu, \Phi)$ does not depend on the observed event times but only on the censoring indicators. Finally, we write the consolidated CSA loss for informative censoring (CSA-INFO) by aggregating (7), (8) and (9) as $\mathcal{L}_{\text{F}}^{\text{CSA-INFO}} \triangleq \mathcal{L}_{\text{F}}^{\text{CSA}} + \ell_c + \ell_{\text{TC}}$.

**Learning**   Model functions $\{h_A(\cdot), \Phi(\cdot), \nu_A(\cdot)\}$ are learned by minimizing the bound (5), via stochastic gradient descent on minibatches from $\mathcal{D}$, with $\mathcal{L}_{\text{F}}^{\text{CSA}}$ for non-informative censoring and $\mathcal{L}_{\text{F}}^{\text{CSA-INFO}}$ for informative censoring. Further, for the IPM regularization loss in (5), we optimize the dual formulation of the *Wasserstein distance*, via the regularized *optimal transport* (Villani, 2008; Cuturi, 2013). Consequently, we only require $\alpha^{-1}\zeta_{h,\Phi}(x, a)$ to be 1-Lipschitz (Shalit et al., 2017) and $\alpha$ is selected by grid search on the validation set using *only* factual data (details below).

## 4   METRICS

We propose a comprehensive evaluation approach that accounts for both factual and causal metrics. Factual survival outcome predictions are evaluated according to standard survival metrics that measure diverse performance characteristics, such as concordance index (C-Index) (Harrell Jr et al., 1984), mean coefficient of variation (COV) and calibration slope (C-slope) (Chapfuwa et al., 2020). See the SM for more details on these metrics. For causal metrics, defined below, we introduce a nonparametric hazard ratio (HR) between treatment outcomes, and adopt the conventional precision in estimation of heterogeneous effect (PEHE) and average treatment effect (ATE) performance metrics (Hill, 2011). Note that PEHE and ATE require ground truth counterfactual event times, which is only possible in (semi-)synthetic data. For HR, we compare our findings with those independently reported in the literature from gold-standard RCT data.

**Nonparametric Hazard Ratio**   In a medical setting, the population hazard ratio $\text{HR}(t)$ between treatment groups is considered informative thus has been widely used in drug development and RCT (Yusuf et al., 2016; Mihaylova et al., 2012). For example, $\text{HR}(t) < 1, > 1$, or $\approx 1$ indicate *population* positive, negative and neutral treatment effects at time $t$, respectively. Moreover, $\text{HR}(t)$ naturally accounts for both censored and non-censored outcomes. Standard approaches for computing $\text{HR}(t)$ rely on the restrictive proportional hazard assumption from CoxPH (Cox, 1972), which is constituted as a semi-parametric linear model $\lambda(t|a) = \lambda_{\text{b}}(t) \exp(a\beta)$. However, the constant covariate (time independent) effect is often violated in practice (see Figure 2b). For CoxPH, the *marginal* HR between treatment and control can be obtained from regression coefficient $\beta$ learned via maximum likelihood without the need for specifying the baseline hazard $\lambda_{\text{b}}(t)$: $\text{HR}_{\text{CoxPH}}(t) = \frac{\lambda(t|a=1)}{\lambda(t|a=0)} = \exp(\beta)$. So motivated, we propose a nonparametric, model-free approach for computing $\text{HR}(t)$, in which we do not assume a parametric form for the event time distribution or the proportional hazard assumption from CoxPH. This approach only relies on samples from the conditional event time density functions, $f(t_1|x)$ and $f(t_0|x)$, via $t_a = h_A(\cdot)$ from (6).

**Definition 1** *We define the nonparametric marginal Hazard Ratio and its approximation,* $\hat{\text{HR}}(t)$, *as*

$$\text{HR}(t) = \frac{\lambda_1(t)}{\lambda_0(t)} = \frac{S_0(t)}{S_1(t)} \cdot \frac{S_1'(t)}{S_0'(t)}, \quad \hat{\text{HR}}(t) = \frac{\hat{S}_0^{\text{PKM}}(t)}{\hat{S}_1^{\text{PKM}}(t)} \cdot \frac{m_1(t)}{m_0(t)}, \quad (10)$$

where for $\text{HR}(t)$ we leveraged (1) to obtain (10) and $S'(t) \triangleq dS(t)/dt$. The nonparametric assumption for $S(t)$ makes the computation of $S'(t)$ challenging. Provided that $S(t)$ is a monotonically decreasing function, for simplicity, we fit a linear function $S(t) = m \cdot t + c$, and set $S'(t) \approx m$. Note that the linear model is *only* used for estimating $S'(t)$ from the nonparametric estimation of $S(t)$.

Table 1: Performance comparisons on ACTG-SYNTHETIC data, with 95% $\text{HR}(t)$ confidence interval. The ground truth, test set, hazard ratio is $\text{HR(t)} = 0.52_{(0.39,0.71)}$.

| Method | Causal | | | Factual | | |
|---|---|---|---|---|---|---|
| | $\epsilon_{\text{PEHE}}$ | $\epsilon_{\text{ATE}}$ | $\text{HR}(t)$ | C-Index (A=0, A=1) | Mean COV | C-Slope (A=0, A=1) |
| CoxPH-Uniform | NA | NA | $0.97_{(0.86,1.09)}$ | NA | NA | NA |
| CoxPH-IPW | NA | NA | $0.48_{(0.03,7.21)}$ | NA | NA | NA |
| CoxPH-OW | NA | NA | $0.60_{(0.53,0.68)}$ | NA | NA | NA |
| Surv-BART | 352.07 | 77.89 | $0.0_{(0.0,0.0)}$ | (0.706, 0.686) | 0.001 | $(0.398, \infty)$ |
| AFT-Weibull | 367.92 | 133.93 | $0.47_{(0.47,0.47)}$ | (0.21, 0.267) | 6.209 | (0.707, 0.729) |
| AFT-log-Normal | 377.76 | 157.64 | $0.47_{(0.47,0.47)}$ | (0.675, 0.556) | 6.971 | (0.707, 0.729) |
| SR | 369.47 | 88.55 | $0.38_{(0.33,0.65)}$ | (0.791, 0.744) | 0 | (0.985, 1.027) |
| CSA (proposed) | 358.72 | **0.8** | $0.45_{(0.39,0.65)}$ | (0.787, 0.767) | 0.131 | (0.985, 1.026) |
| CSA-INFO (proposed) | **344.3** | 31.19 | $\mathbf{0.53}_{(0.41,0.67)}$ | (0.78, 0.764) | 0.13 | (0.999, 1.029) |

Bias from $S'(t)$ can be reduced by considering more complex function approximations for $S(t)$, *e.g.*, polynomial or spline. For the nonparametric estimation of $S(t)$ we leverage the *model-free* population point-estimate-based nonparametric Kaplan-Meier (Kaplan & Meier, 1958) estimator of the survival function $\hat{S}^{\text{PKM}}(t)$ in Chapfuwa et al. (2020) to marginalize both *factual* and *counterfactual* predictions given covariates $x$. The approximated hazard ratio, $\hat{\text{HR}}(t)$, is thus obtained by combining the approximations $\hat{S}_a^{\text{PKM}}(t)$ and $m_a$. A similar formulation for the conditional, $\hat{\text{HR}}(t|x)$, can also be derived. See the SM for full details on the evaluation or $\hat{\text{HR}}(t)$ and $\hat{\text{HR}}(t|x)$. Note that for some AFT- or CoxPH-based parametric formulations, $\text{HR}(t|x)$, can be readily evaluated because $f(t_a|x)$ and $S(t_a|x)$ are available in closed form.

In the experiments, we will use $\text{HR}(t)$ to compare different approaches against results reported in RCTs (see Tables 1 and 3). Further, we will use $\text{HR}(t|x)$ to illustrate *stratified* treatment effects (see Figure 2). Note that though a neural-based survival recommender system (Katzman et al., 2018) has been previously used to estimate $\text{HR}(t|x)$, their approach does not account for confounding or informative censoring thus it is susceptible to bias.

**Precision in Estimation of Heterogeneous Effect (PEHE)**  A general *individualized* estimation error is formulated as $\epsilon_{\text{PEHE}} = \sqrt{\mathbb{E}_X[(\text{ITE}(x) - \hat{\text{ITE}}(x))^2]}$, where $\text{ITE}(x)$ is the ground truth, $\hat{\text{ITE}}(x) = \mathbb{E}_T[\gamma(T_1) - \gamma(T_0)|X = x]$ and $\gamma(\cdot)$ is a deterministic transformation. In our experiments, $\gamma(\cdot)$ is the average over samples from $t_a \sim p_{h,\Phi}(T_A|X = x)$. Alternative estimands, *e.g.*, thresholding survival times $\gamma(T_A) = I\{T_A > \tau\}$, can also be considered as described above.

**Average Treatment Effect (ATE)**  The *population* treatment effect estimation error is defined as $\epsilon_{\text{ATE}} = |\text{ATE} - \hat{\text{ATE}}|$, where $\text{ATE} = \mathbb{E}_X[\text{ITE}(x)]$ (ground truth) and $\hat{\text{ATE}} = \mathbb{E}_X[\hat{\text{ITE}}(x)]$.

## 5 EXPERIMENTS

We describe the baselines and datasets that will be used to evaluate the proposed counterfactual survival analysis methods (CSA and CSA-INFO). Pytorch code including the new semi-synthetic dataset (see below) will be made publicly available. Throughout the experiments, we use the standard $\text{HR}(t)$ for CoxPH based methods and (10) for all others. The bound in (5) is sensitive to $\alpha$, thus we propose approximating *proxy* counterfactual outcomes $\{Y_{\text{CF}}, \delta_{\text{CF}}\}$ for the validation set, according to the covariate Euclidean nearest-neighbour (NN) from the training set. We select the $\alpha$ that minimizes the validation loss $\mathcal{L} = \mathcal{L}_F + \mathcal{L}_{\text{CF}}$ from the set $(0, 0.1, 1, 10, 100)$.

**Baselines**  We consider the following competitive baseline approaches: ($i$) *propensity* weighted CoxPH (Schemper et al., 2009; Buchanan et al., 2014; Rosenbaum & Rubin, 1983); ($ii$) IPM (5) regularized AFT (log-Normal and Weibull) models; ($iii$) an IPM (5) regularized *deterministic* semi-supervised regression (SR) model with accuracy objective from (Chapfuwa et al., 2018), as a contrast for the proposed stochastic predictors (CSA and CSA-INFO); and ($iv$) survival Bayesian additive regression trees (Surv-BART) (Sparapani et al., 2016). For CoxPH, we consider three normalized weighting schemes: ($i$) inverse probability weighting (IPW) (Horvitz & Thompson, 1952; Cao et al., 2009), where $\text{IPW}_i = \frac{a_i}{\hat{e}_i} + \frac{1-a_i}{1-\hat{e}_i}$; $ii$) overlapping weights (OW) (Crump et al., 2006; Li et al., 2018), where $\text{OW}_i = a_i \cdot (1 - \hat{e}_i) + (1 - a_i) \cdot \hat{e}_i$; and $iii$) the standard RCT uniform assumption. A simple linear logistic model $\hat{e}_i = \sigma(x_i; w)$, is used as an approximation, $\hat{e}_i$, to the unknown propensity score $P(A = 1|X = x)$. See the SM for a details of the baselines.

Table 3: Performance comparisons on FRAMINGHAM data, with 95% HR($t$) confidence interval. Test set NN assignment of $y_{CF}$ and $\delta_{CF}$ yields biased HR(t) $= 1.23_{(1.17,1.25)}$, while previous large scale longitudinal RCT studies estimated HR(t) $= 0.75_{(0.64,0.88)}$ (Yusuf et al., 2016).

| Method | Causal | Factual | | |
|---|---|---|---|---|
| | HR($t$) | C-Index (A=0, A=1) | Mean COV | C-Slope (A=0, A=1) |
| CoxPH-Uniform | $1.69_{(1.38,2.07)}$ | NA | NA | NA |
| CoxPH-IPW | $1.09_{(0.76,1.57)}$ | NA | NA | NA |
| CoxPH-OW | $0.88_{(0.73,1.08)}$ | NA | NA | NA |
| Surv-BART | $14.99_{(14.9,14.9e8)}$ | (0.629, 0.630) | 0.003 | (0.232, 0.084) |
| AFT-Weibull | $1.09_{(1.09,1.09)}$ | (0.734, 0.395) | 8.609 | (0.857, 0.89) |
| AFT-log-Normal | $1.55_{(1.46,1.55)}$ | (0.68, 0.56) | 10.415 | (0.979, 0.732) |
| SR | $0.58_{(0.53,0.71)}$ | (0.601, 0.57) | 0 | (0.491, 0.63) |
| CSA (proposed) | $1.04_{(1.00,1.09)}$ | (0.763, 0.728) | 0.161 | (0.891, 0.81) |
| CSA-INFO (proposed) | $\mathbf{0.81}_{(0.77,0.83)}$ | (0.752, 0.651) | 0.156 | (0.907, 0.881) |

**Datasets** We consider the following datasets: ($i$) FRAMINGHAM, is an EHR-based longitudinal cardiovascular cohort study that we use to evaluate the effect of statins on future coronary heart disease outcomes (Benjamin et al., 1994); ($ii$) ACTG, is a longitudinal RCT study comparing monotherapy with Zidovudine or Didanosine with combination therapy in HIV patients (Hammer et al., 1996); and ($iii$) ACTG-

Table 2: Summary statistics of the datasets.

| | FRAMINGHAM | ACTG | ACTG-SYNTHETIC |
|---|---|---|---|
| Events (%) | 26.0 | 26.9 | 48.9 |
| Treatment (%) | 10.4 | 49.5 | 55.9 |
| $N$ | 3,435 | 1,054 | 2,139 |
| $p$ | 32 | 23 | 23 |
| Missing (%) | 0.23 | 1.41 | 1.38 |
| $t_{max}$ (days) | 7,279 | 1,231 | 1,313 |

SYNTHETIC, is a semi-synthetic dataset based on ACTG covariates. We simulate potential outcomes according to a Gompertz-Cox distribution (Bender et al., 2005) with selection bias from a simple logistic model for $P(A = 1|X = x)$ and AFT-based censoring mechanism. The generative process is detailed in the SM. Table 2 summarizes the datasets according to ($i$) covariates of size $p$; ($ii$) proportion of non-censored events, treated units, and missing entries in the $N \times p$ covariate matrix; and ($iii$) time range $t_{max}$ for both censored and non-censored events. Missing entries are imputed with median or mode if continuous or categorical, respectively.

**Quantitative Results** Experimental results for two data-sets in Tables 1 and 3, illustrate that AFT-based methods are high variance, inferior in calibration and C-Index than accuracy-based methods (SR, CSA, CSA-INFO). Surv-BART is the least calibrated but low variance method. CSA-INFO and CSA outperform all methods across all factual metrics, whereas CSA-INFO is better calibrated, low variance but slightly lower C-Index than CSA. Note that we fit CoxPH using the entire dataset; since it does not support counterfactual inference, we do not present factual metrics. By properly adjusting for both informative censoring and selection bias, CSA-INFO significantly outperforms all methods in treatment effect estimation according to HR($t$) and $\epsilon_{PEHE}$, across non-RCT datasets, while remaining comparable to AFT-Weibull on the RCT dataset (see the SM). Further, RCT-based results on ACTG data in the SM illustrate comparable HR($t$) across all models except for AFT-log-Normal and Surv-BART, which overestimate, and SR, which underestimates risk. For non-RCT datasets (ACTG-SYNTHETIC and FRAMINGHAM), CoxPH-OW has a clear advantage over all CoxPH based methods, mostly credited to the well-behaved bounded propensity weights $\in [0, 1]$. Interestingly, the FRAMINGHAM observational data exhibits a common paradox, where without proper adjustment of selection and censoring bias, naive approaches would result in a counter-intuitive treatment effect from statins. However, there is severe *confounding* from covariates such as age, BMI, diabetes, CAD, PAD, MI, stroke, *etc.*, that influence both treatment likelihood and survival time. Table 3, demonstrates that CSA-INFO is clearly the best performing approach. Specifically, its HR($t$), reverses the biased observational treatment effect, to demonstrate positive treatment from statins, which is consistent with prior large RCT longitudinal findings (Yusuf et al., 2016).

**Qualitative Results** Figure 2a demonstrates that CSA-INFO matches the ground truth population hazard, HR($t$), better than alternative methods on ACTG-SYNTHETIC data. See the SM for ACTG and FRAMINGHAM. Figure 2b shows sub-population log hazard ratios for four patient clusters obtained via hierarchical clustering on the individual log hazard ratios, $\log \text{HR}(t|x)$, of the test set of FRAMINGHAM data. Interestingly, these clusters stratify treatment effects into: positive (2), negative (1 and 3), and neutral (4) sub-populations. Moreover, the estimated density of median $\log \text{HR}(t|x)$ values in Figure 2c illustrates that nearly 70% of the testing set individuals have $\log \text{HR}(t|x) < 0$, thus may benefit from taking statins. Further, we isolated the extreme top and bottom quantiles, $\text{HR}(t|x) < 0.024$ and $\text{HR}(t|x) > 1.916$, respectively, of the median $\log \text{HR}(t|x)$ values for the

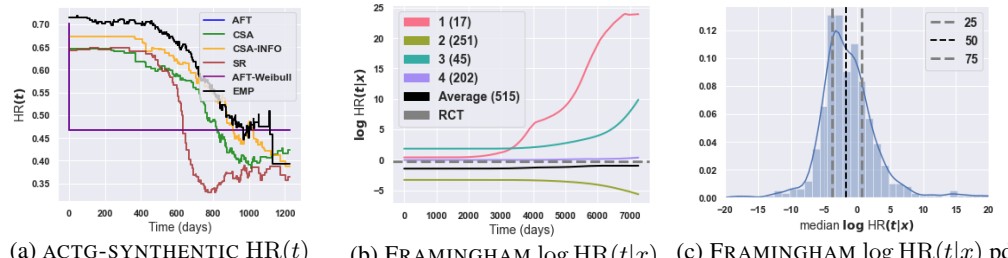

(a) ACTG-SYNTHENTIC HR$(t)$  (b) FRAMINGHAM log HR$(t|x)$  (c) FRAMINGHAM log HR$(t|x)$ pdf

Figure 2: (a) Inferred population HR$(t)$ compared against ground truth (EMP) on ACTG-SYNTHETIC data. CSA-INFO-based (b) cluster-specific average log HR$(t|x)$ curves and (c) estimated density of median log HR$(t|x)$ values on the test set of the FRAMINGHAM dataset. Clusters assignment were obtained via hierarchical clustering of individualized log HR$(t|x)$ traces.

test set of FRAMINGHAM, as shown in Figure 2c. After comparing their covariates, we found that individuals with the following characteristics may benefit from taking statins: young, male, diabetic, without prior history (CAD, PAD, stroke or MI), high BMI, cholesterol, triglycerides, fasting glucose, and low high-density lipoprotein. There seem to be consensus that diabetics and high-cholesterol patients benefit from statins (Cheung et al., 2004; Wilt et al., 2004). See SM for additional results.

## 6 CONCLUSIONS

We have proposed a unified counterfactual inference framework for survival analysis. Our approach adjusts for bias from two unknown sources, namely, *confounding* due to covariate dependent selection bias and *censoring* (informative or non-informative). Relative to competitive alternatives, we demonstrate superior performance for both survival-outcome prediction and treatment-effect estimation, across three diverse datasets, including a semi-synthetic dataset which we introduce. Moreover, we formulate a model-free nonparametric hazard ratio metric for comparing treatment effects or leveraging prior randomized real-world experiments in longitudinal studies.

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
