# OpenReview forum: "Enabling counterfactual survival analysis with balanced representations"
_ICLR.cc/2021/Conference — Reject_

### Official Review · AnonReviewer4 · 2020-10-23
**Well-articulated paper extending individualized treatment effect estimation to time-to-event outcomes**

**Rating:** 7
**Confidence:** 4

**Review:**

This paper is very well written. The motivation and formalism is also clear with every step in the argument properly justified. The extension of individualized treatment effects to survival data is in some sense straightforward as both areas are quite mature and can be unified with aggregated loss functions dealing with biases of different type. The proposed solution, metrics and datasets proposed for this problem are compelling though and I believe will serve as a benchmark for further studies on treatment effects and survival data.

One question I have is on Corollary 1. I don't see a meaningful difference between this statement and that given by (Shalit et al, 2017), nor a proof in the Appendix. Why the separate statement?

The experiments are somewhat underwhelming. Survival BART has also been developed for treatment effect estimation [1]. This benchmark is certainly more relevant. Similarly, Survival-based deep learning architectures have been developed which could have been considered as well [2]. At least considering these, perhaps modelling each treatment group separately or including treatment as an additional feature should be considered to understand where the source of gain comes from. As presented, since almost all benchmarks consider linear interaction between features I would guess that improvements come from non-linear modelling rather than bias reduction from censoring and selection bias.

[1] Hu, Liangyuan, Jiayi Ji, and Fan Li. "Estimating Heterogeneous Survival Treatment Effect via Machine/Deep Learning Methods in Observational Studies." arXiv preprint arXiv:2008.07044 (2020).
[2] Lee, Changhee, et al. "DeepHit: A Deep Learning Approach to Survival Analysis With Competing Risks." AAAI. 2018.

---

> ### Author Response · Authors · 2020-11-13
> **Response to Reviewer 4**
>
> Thanks for the encouraging comments.
>
> **Ablation study**
>
> We do not compare with either $(i)$ modeling each treatment group separately or $(ii)$ including treatment as an additional feature. Experimental findings from  Shalit et al., 2017, demonstrate that the IPM regularized bound (5) outperforms these approaches in accounting for selection bias.
>
> In the experiments, we compare to competitive IPM regularized approaches parameterized by neural networks such as AFT-Weibull, AFT-log-Normal, and SR. Moreover, we compare to Surv-BART (Sparapani et al., 2016) a Bayesian additive regression trees method and weighted CoxPH methods. Consequently, our experiments are comprehensive and we are confident that the performance benefits are attributed to $(i)$ accounting for informative censoring bias $(ii)$ accounting for selection bias (optimal IPM regularizer with $\alpha > 0$) and $(iii)$  *flexible and non-parametric* generative modeling of event times from the stochastic planar flow.
>
> **Additional baselines**
>
> Survival-specific related works (*e.g.*, `` "Lee, Changhee, et al. DeepHit: A Deep Learning Approach to Survival Analysis With Competing Risks. AAAI. 2018") that do not focus on causal survival analysis were excluded from our discussion as they are out of scope.
>
> Moreover, these methods require significant modifications (like the ones we are proposing) for counterfactual survival analysis to work. First, they require a mechanism for accounting for both selection bias and informative censoring bias. Second, methods that discretize event times, $e.g.$, Lee, Changhee, et al., and Miscouridou et al., may introduce additional biases when computing ${\rm HR} (t)$ or ${\rm HR} (t|x)$, that need to be properly addressed.
>
> We have included the work ``"Hu, Liangyuan, Jiayi Ji, and Fan Li. Estimating Heterogeneous Survival Treatment Effect via Machine/Deep Learning Methods in Observational Studies. arXiv preprint arXiv:2008.07044 (2020)" in the introduction. Note that our experiments include Surv-BART (Sparapani et al., 2016) a Bayesian additive regression trees method.
>
> **Loss bound Corollary 1**
>
> For completeness and readability, we write the loss bound (5) from Shalit et al., 2017, in the main paper as it is crucial in our discussion on repurposing it for generative counterfactual survival analysis.

---

### Official Review · AnonReviewer1 · 2020-10-27
**Concerned about the originality and significance; includes many inaccurate statements.**

**Rating:** 4
**Confidence:** 5

**Review:**

Summary:
This paper provides an approach for causal inference in observational survival dataset in which the outcome is of time-to-event type with right-censored samples. The method consists of a representation learning component to reduce selection bias and a survival analysis component which is modeled with normalizing flows.

Pros:
- The paper addresses an important and interesting question.
- The manuscript is well-written and easy to understand.

Cons:
- My main concern is the minimal originality and significance of this work. That is, the representation learning component is directly taken from (Shalit et al., 2017) and the objective function for the survival analysis component is directly taken from (Chapfuwa et al., 2018). Also, the use of normalizing flows for modelling time-to-event kind of targets is not well motivated.
- The literature review only points to several publications but does not go in depth into why/how the proposed method differs from those. There are also some important references that are missing. For example, Miscouridou et al. (2018) also use normalizing flows for survival analysis and it is necessary that the authors discuss how their work differs from theirs.
Miscouridou, X., Perotte, A., Elhadad, N., & Ranganath, R. (2018). Deep survival analysis: Nonparametrics and missingness. In Machine Learning for Healthcare Conference
- the paper suffers from many inaccurate statements; a few examples follow:
1. In the third paragraph of the Introduction section, the authors mention that “the treatment assignment mechanism is not known a priori. Therefore, there may be variables, known as confounders, affecting both the treatment and survival time, which lead to selection bias”. This is wrong; even if we know the treatment assignment policy a priori, we still might have selection bias … these two are independent.
2. In the last sentence of paragraph four of Introduction, the authors say that the methods (cited above) that account for confounding bias (by re-weighting) lack a counterfactual prediction mechanism. This is wrong because trivially, all methods have a prediction mechanism in place, and the ones that do account for confounding bias can predict counterfactuals accurately as well.
3. The authors should note that representation learning does not **remove** confounding bias; it might only **reduce** it. Also, reweighting does not **remove** confounding bias either; it just **accounts for** it.

Minor:
- In section 2, under *Estimands of Interest*, the authors state that “$\lambda(t | x)$ is defined below” but it’s never defined.
- In section 3, under *Accounting for selection bias*, the authors state that we can go from Eq. (1) to (2) because identifiability holds (since “$X$ is a sufficient set from $A$ into $T$.”). Could you please explain how this is different from Ignorability? I.e., $\{ T_0, T1 \} \perp A | X$ ?

---

> ### Author Response · Authors · 2020-11-13
> **Response to Reviewer 1**
>
> Thanks for the insightful comments. On $(i)$ originality and significance and  $(ii)$ planar flow, please refer to the overall comments above.
>
> **Additional references**
>
> Unlike "``Miscouridou et al., Deep survival analysis: Nonparametrics and missingness"  our work formulates a counterfactual survival analysis framework that accounts for  *informative or non-informative* censoring mechanisms and confounding while modeling event times as a continuous variable instead of discretizing time. We have clarified this in our submission.
>
> Would the reviewer point out additional missing references so that we can include them in our discussion? Thanks.
>
> **Unclear statements in the introduction**
>
> We agree with the reviewer in that unknown treatment assignment itself does not lead to selection bias. In the introduction, we wish to emphasize that, in an observational study, the treatment assignment is not known in priori as in randomized trials and it is often the case that treatment depends on some variables, known as confounders, which leads to the selection bias. We apologize for the logical inaccuracy of connecting the unknown treatment assignment with selection bias.
>
> We do not completely agree with this reviewer's statement:  "all methods have a prediction mechanism in place, and the ones that do account for confounding bias can predict counterfactuals accurately as well". Note that though re-weighted CoxPH (or adjusted survival curves) methods account for confounding bias, CoxPH (or adjusted survival curves) lack the counterfactual prediction mechanism, more precisely, the capability to impute the counterfactual outcome for each individual. Specifically, they are built only for estimating treatment effects on average not for predicting individual counterfactual outcomes. Interestingly, since our approach accounts for confounding bias in a counterfactual prediction context, it can be more readily adapted to other prediction mechanisms for event times.
>
> We agree in that representation learning does not remove confounding bias, we have replaced "remove" with "account" on page 2, P2.
>
> **Minor comments**
>
> We apologize for the confusion:
>
> $(a)$ $\lambda(t|x)$ is indeed  defined at the end of the paragraph in terms of $f(t|x)\triangleq P(T=t|X=x) = \lambda(t|x)S(t|x)$. We have made this  definition explicit in our submission.
>
> $(b)$ The potential outcome distribution is *identifiable* as a consequence of ignorability (consistent with assumed causal graphs in Figures 1b and 1c), overlap, and consistency. We have updated the language in the paper to clarify this.

---

> > ### Comment · AnonReviewer1 · 2020-11-24
> > **Thank you for the responses.**
> >
> > I would like to thank the authors for their rebuttal.
> >
> > I appreciate the clarifications regarding “Originality and significance” of the work under the “Overall comments” in the authors’ rebuttal, in which **accounting for informative censoring** is stated to be the main contribution of this submission. This, according to the paper, is handled through the use of Equations (8) and (9). These two equations are, however, mirrors of Equation (6), which was first proposed in (Chapfuwa et al., 2018). My evaluation is that this level of contribution does not pass the acceptance threshold. I have updated my score accordingly.
> >
> > Moreover, I still defend my comment *“[the methods that] account for confounding bias can predict counterfactuals accurately as well.”* Assume for example, the re-weighed CoxPH model: we can train one model for the treated group and another for the control group. Since selection bias has been accounted for in learning both models, we can use them to estimate the counterfactuals. Additionally, the following statement in the rebuttal is wrong: *“[CoxPH is] built only for estimating treatment effects on average not for predicting individual counterfactual outcomes.”* This is because CoxPH does take the covariates into account for deriving the hazard function, so indeed, CoxPH is an patient-specific survival analysis method.

---

> > > ### Author Response · Authors · 2020-11-25
> > > **On contributions and counterfactual survival analysis**
> > >
> > > We want to thank the reviewer for engaging during the discussion period.
> > >
> > > **Counterfactual survival analysis**
> > >
> > > We want to clarify that counterfactual inference in the context of survival analysis refers to **time-to-event** predictions of potential outcomes $(T_1, T_0)$, conditioned on the covariates $X$.
> > > As detailed in the supplementary material in equation (12) (also see [1] for additional details), the re-weighted CoxPH estimates the treatment effect as the coefficient of a CoxPH model in which the treatment indicator is used as the input to the model and the likelihood is weighted by (propensity-based) weights,  *e.g.* inverse propensity weights.
> > > As a result, the estimated coefficient of the model, $\beta$, is the treatment effect estimate.
> > > However, since in the CoxPH model the inputs to the model are not the covariates, $X$, but the treatment indicator, the model cannot be used to make individual-level predictions.
> > > In other words, as we pointed out in the rebuttal, CoxPH is built only for estimating the treatment effect (on average), not for predicting individual counterfactual outcomes.
> > > Moreover, since in a re-weighted CoxPH model the input to the model is the treatment indicator, there is no point in building one model for the treated and one for the non-treated as the reviewer suggests.
> > >
> > > [1] A. Buchanan, et al., "Worth the weight: using inverse probability weighted cox
> > > models in aids research". AIDS research and human retroviruses, 2014
> > >
> > > **Contributions**
> > >
> > > First, we want to clarify that accounting for informative-censoring is  **one** of our main contributions, other contributions are also detailed in the **overall response** above.
> > >
> > > Though (8) and (9) mirror (6) they are *not* equivalent. While the modification may seem simple according to the reviewers' assessment, it is surprising that such a modification has *not* been applied to counterfactual survival analysis (or survival analysis predictions). Importantly, extensive experimental results presented in the paper indicate that CSA-INFO consistently outperforms competitive baselines in treatment effect estimation and survival outcome prediction (per the metrics: mean COV, Calibration, and C-Index) in both real-world and synthetic datasets.
> > >
> > > Also, key to our contributions is the novel *model-free* hazard ratio (HR) estimator, allowing us to: $(i)$ evaluate our predictions against *real-world* randomized experiments on the FRAMINGHAM dataset (Table 3), and $(ii)$ identify heterogeneous treatment effects of statins according to the stratified ${\rm HR}(t|x)$ (Figure 2).
> > >
> > > Finally, we introduce a comprehensive framework for counterfactual survival analysis leveraging balanced representation learning, where none existed with $(i)$ survival-specific metrics, $(ii)$ a new survival-specific semi-synthetic dataset, and $(iii)$ an evaluation approach for comparing with real-world randomized experiments instead of solely relying on synthetic datasets.

---

### Official Review · AnonReviewer2 · 2020-10-29
**Under the survival analyses setting, this paper concentrates on counterfactual inference related to individualized treatment effect, especially hazard ratio. Bound proposed in Shalit et al., 2017 is adopted for model learning by minimizing the upper bound which consists of factual loss and integral probability metric. Proposed factual loss is similar to Chapfuwa et al. (2018) for non-informative censoring case, and adding extra loss terms for informative censoring case. Simulation result shown.**

**Rating:** 7
**Confidence:** 4

**Review:**

Disclosure: I found this paper online during review process https://arxiv.org/abs/2006.07756


This is a comprehensive paper with interesting application of counterfactual inference under survival analysis setting. Overall, my recommendation is to accept.
•	It nice that the proposed nonparametric approach in this paper can adjusts for bias from confounding due to covariate dependent selection bias and censoring (informative or non-informative).
•	Under three criterions [concordance index (C-Index) (Harrell Jr et al., 1984), mean coefficient of variation (COV) and calibration slope (C-slope) (Chapfuwa et al., 2019)] and three datasets [FRAMINGHAM, ACTG, semi synthetic ACTG], compared proposed method with 7 seven others, including survival Bayesian additive regression trees (Surv-BART) (Sparapani et al., 2016) [using nonparametric Kaplan-Meier based estimator] and Cox proportional hazard model (using real HR form, using three normalized weighting schemes).
•	P6, equation (10), the nonparametric form is a natural adoption of KM estimator. We know S^{‘}=-f(t). Wondering the motivation of choosing a linear approximation to S, and curious would the cardiovascular and HIV data adopted happen to be with S not so curved? Could you shed light on these?
•	P3, assumption of “no unobserved confounders or ignorability” sounds strong. Understand the mathematical challenge if relaxing it. Maybe for future research.
•	The overall presentation is nice. The organization of a few places might be improved to help first time reader to follow, eg. ITE initially defined on P3 without example till two paragraphs below, h_{A} first mentioned with no prior definition and no explicit math relation to p(T|X), briefly specify “Do (A=a)” is for effect of intervention.
•	Minor issues like align the symbols used across the paper, eg. add subscript when define S^{‘}_{i},  m_{i}, ... i=0, 1 to increase clarity.

---

> ### Author Response · Authors · 2020-11-13
> **Response to Reviewer 2**
>
> Thanks for the encouraging comments.
>
> **Nonparametric estimation of population ${\rm HR} (t)$ or individualized ${\rm HR} (t|x)$ hazard ratio**
>
> Provided that $S(t)$ is a monotonically decreasing function, for simplicity, we fit a linear function $S(t) = m \cdot t + c$, and set $S^{\prime}(t) \approx m$. Fortunately, $S(t)$ in both the  FRAMINGHAM (cardiovascular) and ACTG (HIV) datasets follows an approximately linear function with a slight curve toward $t_{\rm max}$. Note that the linear model is *only* used for estimating $S^{\prime}(t)$ from a nonparametrically estimated $S(t)$. To reduce approximation error, more complex functions, *e.g.*, polynomial or spline may be considered. Such exploration is left as future work.
>
> **No unobserved confounders or ignorability**
>
> We completely agree in that ignorability is a strong assumption, however, relaxing it (*e.g.*, understanding the sensitivity of our estimates to unobserved confounding) is a hard problem, thus is left as future work.
>
> **Minor Issues**
>
> For readability, we have pointed to Section 2  "Estimands of Interest" when initially introducing ITE in P3 and improved the language on potential outcome estimation.

---

### Official Review · AnonReviewer3 · 2020-10-29
**Important problem but limited methodological contribution.**

**Rating:** 5
**Confidence:** 4

**Review:**

**Summary and key claims**

This paper repurposes the balanced representation learning framework for estimating treatment effects, originally proposed in (Shalit et al., 2017), for the survival prediction setup. The proposed model deals with selection bias induced by confounded treatment variables, and in addition, deals with censoring bias and informative censoring.

*The key contributions claimed by the paper are:*
- Developing a loss function incorporating adjustments for informative censoring and selection bias.
- Developing a generative model for event times based on planar normalizing flows.
- Proposal of survival-specific evaluation metrics, including a new nonparametric hazard ratio estimator.

**Originality and significance**

Overall, I think that the paper is a straightforward application of the balanced representation method in (Shalit et al., 2017) to survival outcomes. It does not seem like survival prediction is any different from the conventional ITE setup with respect to the representation learning aspect of the model, hence I don't think that the paper contributes methodologically to the problem of handling selection bias. Moreover, the censoring terms in the loss function are also very similar to those introduced previously in Chapfuwa et al ICML 2018 paper. Based on this, I think that the extent of technical contribution in the paper does not pass the acceptance threshold.

I was expecting some more analysis on the interplay between censoring bias, selection bias and the effect of both on causal identifiability. Unfortunately, it seems that authors chose to address each of these impairments/biases separately using existing solutions, and the resulting model is simply an amalgamation of existing ideas.

The "generative modeling" part of the model is somehow alien to the original problem of estimating treatment effects on survival outcomes. It is not clear why planar flows were specifically used and why flows are needed at all since the complexity needed in modeling is in the relation between features and survival parameters and not the complexity of the survival distribution itself. The usage of normalizing flows on the output layer is fine but seems to me unnecessary; this reinforces my impression of the model being all over the place.

On the positive side, I think that the problem of estimating ITEs on survival is very important and rarely addressed. Most ML models for ITEs focus on real-valued targets, but this is rarely the relevant setup in practice as survival is often the measure of treatment efficacy in medicine. I also think that the idea of comparing the estimated HRs of RCTs with the ones recovered by the model is a very smart way to evaluate counterfactual inferences, and can be a useful evaluation metric for future papers.

---

> ### Author Response · Authors · 2020-11-13
> **Response to Reviewer 3**
>
> Thanks for the insightful comments. On $(i)$ originality and significance and $(ii)$  planar flow, please refer to the overall comments above.
>
> **Censoring bias and selection bias**
>
> Understanding the effect of both censoring bias and selection bias on causal identifiability is interesting and challenging, thus left as future work.
>
> Note that the IPM regularizer accounting for selection bias, from bound in (5) is sensitive to $\alpha$, thus we propose approximating proxy counterfactual outcomes $\{Y_{\rm CF}, \delta_{\rm CF} \}$ for the validation set, according to the covariate Euclidean nearest-neighbour (NN) from the training set. We select the $\alpha$ that minimizes the validation loss $L = L_{\rm F} + L_{\rm CF}$ from the set $(0, 0.1, 1, 10, 100)$.
>
> Moreover, experimental findings demonstrate that accounting for informative censoring is key, as CSA-INFO consistently outperforms CSA in causal metrics.

---

### Author Response · Authors · 2020-11-13
**Overall comments**

We thank the reviewers for the encouraging and insightful comments, below are the point-by-point responses and summary of changes (also highlighted in the updated revision).

**Originality and significance**

First, we want to clarify that neither Shalit et al., 2017 nor  Chapfuwa et al., 2018,  present a framework directly applicable to counterfactual survival analysis.

Though we leverage the representation learning approach in Shalit et al., 2017, which only accounts for selection bias, our contributions are significant and non-trivial. We demonstrate that we still need to account for censoring (non-informative and informative). Chapfuwa et al., 2018 (and other deep learning survival analysis models), only addresses non-informative censoring, it lacks a counterfactual inference framework, and it does not account for selection bias.

We demonstrate the importance of censoring and that accounting for informative censoring is key to counterfactual survival analysis. We model informative censoring as a separate potential outcomes distributions for censoring times and introduce new losses that account for informative censoring biases. Moreover, current deep learning survival analysis methods neglect informative censoring and require significant modifications for counterfactual survival analysis.
Importantly, we present extensive experimental findings (Tables 1 and 3) demonstrating that, by accounting for both informative censoring and selection bias, CSA-INFO consistently outperforms competitive baselines in treatment effect estimation and survival outcome prediction (per the metrics: mean COV, Calibration, and C-Index) in both real-world and synthetic datasets.

We agree with R3 in that future work in treatment effect estimation and model evaluation will benefit from comparisons with the reported hazard ratio (HR), ${\rm HR} (t)$, from prior large scale *real-world* RCTs, as we have shown in Table 3, instead of solely resorting to commonly used synthetic datasets. We enable this important model evaluation approach with the proposed  *model-free*  ${\rm HR} (t)$ estimator, which is also a contribution of our work. Further, we use the estimated ${\rm HR}(t|x)$ to illustrate its use for stratified treatment effects (see Figure 2). The proposed HR approach is an important improvement over the widely used CoxPH (see Figure 2) which assumes constant population hazard ${\rm HR} (t)$ over time. Moreover, under CoxPH, ${\rm HR} (t|x)$ cannot be computed as in the way it is widely used in practice, it  *lacks* a counterfactual inference mechanism.

As pointed out by R2 and R4, we present a comprehensive framework for counterfactual survival analysis. Specifically, survival-specific metrics, a new survival-specific semi-synthetic dataset, and demonstrate an approach for leveraging prior RCTs in *real-world* treatment effect model evaluation using the proposed *model-free* ${\rm HR} (t)$ estimator. Finally, this work will serve as an important baseline for future work in real-world counterfactual survival analysis.

**Planar flow**

We want to clarify that we are *not* using a normalization flow likelihood in the output of the model. Instead, we leverage the planar flow formulation from Rezende  \& Mohamed, 2015,  *only* to introduce stochasticity and thus being able to synthesize (sample) event time predictions from a nonparametric $t \sim P(T_A | X=x)$. Note that we can in principle use an arbitrary nonlinear transformation of a simple random distribution, thus we do not consider the use of the planar flow a methodological contribution of our work. We have clarified this important point in the revision.

Importantly, in the experiments we demonstrate the benefits of stochastic predictors (CSA and CSA-INFO), not necessarily the use of the planar flow, by contrasting them to the IPM-regularized *deterministic* semi-supervised regression (SR) model with accuracy objective from (7), as shown in Tables 1 and 3.

Moreover, the neural-network-based generative model for event times relaxes restrictive survival linear and parametric assumptions (*e.g* AFT-based), thus allowing for more flexible modeling. Further, the model can also produce nonparametric uncertainty estimates for ITE predictions.

---

### Decision · Program_Chairs · 2021-01-07
**Final Decision**

**Decision:**

Reject

**Comment:**

Summary: This paper provides an approach for causal inference in observational survival dataset in which the outcome is of time-to-event type with right-censored samples.  To this end, the paper adapts the balanced representation learning approach proposed in (Shalit et al, 2017) to the context of survival analysis.
The paper adapts an approach that uses flexible models to learn nuisance models, common in machine learning.

The authors validated their approach via simulation study and a set of application datasets: a EHR-based cohort study of cardio-vascular health, an RCT dataset of HIV patients, and a semi-synthetic dataset.

The main concerns of reviewers were due to perceived lack of originality relative to the original proposal in (Shalit et al, 2017)